# The Immunomodulatory Effect of Different FLT3 Inhibitors on Dendritic Cells

**DOI:** 10.3390/cancers16213719

**Published:** 2024-11-04

**Authors:** Sebastian Schlaweck, Alea Radcke, Sascha Kampmann, Benjamin V. Becker, Peter Brossart, Annkristin Heine

**Affiliations:** 1Medical Clinic III for Hematology, Oncology, Rheumatology, Immunoncology and Stem-Cell Transplantation, University of Bonn, 53127 Bonn, Germany; sebastian.schlaweck@ukbonn.de (S.S.); alea.radcke@ukbonn.de (A.R.); sascha.kampmann@ukbonn.de (S.K.); peter.brossart@ukbonn.de (P.B.); 2Faculty of Medicine, Mildred Scheel School of Oncology Aachen Bonn Cologne Düsseldorf (MSSO ABCD), University Hospital of Bonn, 53127 Bonn, Germany; 3Department of Radiology and Neuroradiology, Bundeswehr Central Hospital, 56072 Koblenz, Germany; benjamin3becker@bundeswehr.org

**Keywords:** AML, FLT3, dendritic cells, gilteritinib, midostaurin, quizartinib

## Abstract

Approximately 30% of acute myeloid leukemias are genetically defined by alteration in the FMS-like tyrosine kinase 3 (FLT3). Tyrosine kinase inhibitors targeting FLT3 improved the prognosis of patients with AML but also increased infection rates. Dendritic cells are professional antigen-presenting cells inducing robust immune responses. Therefore, we investigated the immunomodulatory effect of three different FLT3 inhibitors, midostaurin, gilteritinib and quizartinib, on normal dendritic cells in vitro. In this study, we unveil the immunosuppressive effect of midostaurin on DCs by flow cytometry, RNA sequencing and western blotting. The effect of gilteritinib was less pronounced, while quizartinib exerted almost no immunosuppression. Our data may provide an explanation for increased infection rates observed in clinical trials and will help to choose the optimal compound for therapy.

## 1. Introduction

FMS-like tyrosine kinase 3 (FLT3) is a tyrosine kinase essential in regulating survival, proliferation, and differentiation in hematopoietic stem cells [1].

Mutations in the tyrosine kinase domain (TKD) or internal tandem duplication in the juxtamembrane domain (ITD) region lead to ligand-independent activation and consequent anti-apoptotic and proliferative effects in leukemic stem cells by activation of multiple signaling pathways, including STAT5, MAPK, and AKT [2].

In acute myeloid leukemia (AML), FLT3 has become a promising therapeutic target, as FLT3 TKD- and ITD-positive AML account for almost one-third of all cases [3,4]. FTL3 alterations (FLT3_pos_) are associated with impaired survival [5] and drugs directly targeting FLT3_pos_ are approved for AML treatment.

The RATIFY study has shown the promising impact of the first-generation FTL3 inhibitor (FLT3i) midostaurin in the primary treatment of FLT3_pos_ AML [6]. Additionally, the novel second-generation TKI gilteritinib has also been shown to be active in the relapsed/refractory setting [7]. To prevent relapse, allogeneic hematopoietic stem cell transplantation (alloHSCT) is a therapeutic option for transplant-eligible patients with FLT3_pos_ AML after induction therapy. Of note, targeting FLT3 as maintenance therapy in AML patients after alloHSCT with sorafenib reduced relapse rates [8,9] but caused relevant side effects [10]. The RADIUS trial investigated the effect of midostaurin as maintenance therapy after alloHSCT and showed decreased relapse rates after FLT3-directed therapy post-alloHSCT [11]. However, taking into account that FLT3-directed first-line therapy during induction was not standard therapy when investigating the effect of sorafenib after alloHSCT, the optimal treatment/maintenance therapy after alloHSCT for FLT3_pos_ AML is not set.

Recent data add another piece of knowledge as the QuANTUM trial has proven the efficacy of quizartinib, another second-generation TKI, for induction and maintenance treatment in FLT3-ITD AML [12].

Previous research has impressively shown that TKIs modulate immune responses [13,14]. Previous research has shown that midostaurin and gilteritinib are multitargeted protein kinases [15,16], while quizartinib shows exceptional kinase selectivity [17]. Especially in the context of alloHSCT, the magnitude of immune modulation is of great interest as (i) the risk for infections is higher, (ii) allogeneic immune cells are likely to cause Graft versus Host disease (GvHD), a potentially life-threatening complication after alloHSCT [18] and (iii) proper immune function is mandatory for immunological surveillance to maintain disease control. Interestingly, stimulation of the FLT3/FLT3 ligand pathway prevents GvHD in murine studies [19].

A central player in initiating immune responses are dendritic cells (DCs). DCs are professional antigen-presenting cells and induce robust T-cell responses and T-cell activation [20]. In DCs, FLT3/FLT3 ligand signaling is essential for their development. Moreover, stimulation of the FLT3 signaling enhances DC responses in the context of vaccinations and malignancies [21]. FLT3 ligand treatment not only increases the number of DCs but also induces regulatory T cells in a DC-dependent manner [19].

Essential for this process is the regulation of co-stimulatory B7 molecules [22] on the DC surface to interact with T cells and the secretion of cytokines, such as IL-12 [23], to modulate immune responses.

Therefore, we investigated the immunomodulatory effect on dendritic cells of three FLT3i: midostaurin, gilteritinib and quizartinib.

## 2. Methods

### 2.1. Generation of Bone Marrow-Derived Dendritic Cells

To generate bone marrow-derived murine dendritic cells (bmDCs), bone marrow cells from C57BL/6J mice were isolated and differentiated in the presence of GM-CSF, as described previously [13]. Briefly, the medium was changed every second day, and the different FLT3i, midostaurin, gilteritinib and quizartinib (all purchased from Selleckchem, Houston, TX, USA), were added at specific concentrations ranging from 1 to 100 nM every other day for the whole differentiation period of 6 days to mimic clinical routine, where continuous TKI treatment affects DC differentiation and development as well. After 6 days of culture, bmDCs were stimulated with the TLR4 ligand lipopolysaccharide (LPS) in a concentration of 100 ng/mL. Cells were used for further experiments on day 7.

For generation of human dendritic cells, an established protocol was used [14]. Briefly, GM-CSF and IL-4 were added to the medium every other day to generate human monocyte-derived dendritic cells (moDCs) and, when indicated, one of the three FLT3i at the most potent concentration of 100 nM.

### 2.2. Flow Cytometry

Murine and human DCs were stained with commercially available fluorophore-conjugated monoclonal antibodies and measured at a BD FACSCantoII (BD Biosciences, East Rutherford, NJ, USA) or Sony ID7000 (Sony Biotechnology, San Jose, CA, USA).

### 2.3. Cytokine Measurement

Supernatants of bmDC cultures were harvested after 18 h of LPS stimulation. The cytokine concentration was determined using the LEGENDplex™ Mouse Inflammation Panel (13-plex) kit or ELISA MAX Deluxe kits (both from Biolegend, San Diego, CA, USA) according to the manufacturer’s instructions.

### 2.4. RNA Isolation

RNA was harvested from moDCs on day 7 using the Quick-RNA™ Microprep Kit (Zymo Research, Irvine, CA, USA), according to the manufacturer’s instructions. RNA quality was controlled using a Tapestation 4200 (Agilent Technologies, Santa Clara, CA, USA).

### 2.5. The 3′-mRNA Sequencing

The libraries were prepared using the QuantSeq FWD 3′-mRNA-Seq Kit from Lexogen, Wien, Austria. Sequencing was performed on the NovaSeq 6000 platform (Illumina, San Diego, CA, USA) with 1 × 100 bp single-end reads following the manufacturer’s protocols and an average of 10 M raw reads per sample.

### 2.6. Analysis of RNA Sequencing Data

For statistical analysis of RNA sequencing data, we followed the workflow already described elsewhere [24]. In brief, FastQ files were aligned to reference genome mm10 and subsequently indexed. For differential gene expression analysis, we followed the workflow described by Love et al. [25] using the default settings of DESeq2 (version 1.42.0) [26] in R (version 4.3.0). Genes were assumed to be differentially expressed with log2 fold changes in FPKM >1 (fragments per kilobase per million mapped fragments) and FDR <0.1. Overlapping and non-overlapping differentially expressed genes between different treatments and dendritic cells were depicted using the web-based tool InteractiVenn (https://www.interactivenn.net/, last accessed 29 September 2024) [27]. Gene set enrichment analysis was performed with ClusterProfiler in R version 4.1.2, as described elsewhere [28]. Over-representation of biological processes and pathways within networks was analyzed with the differentially expressed genes as seeds. For this purpose, protein–protein interactions of these genes were retrieved from the Innate database [29] and analyzed using the web-based tool NetworkAnalyst (https://www.networkanalyst.ca/, last accessed 29 September 2024) [30].

### 2.7. Western Blot

As described earlier [31], first, whole cell lysates were generated, and after measurement of the protein concentration, 20 µg were separated on a polyacrylamide gel and transferred onto a nitrocellulose membrane. For the detection of protein bands, an enhanced chemiluminescence kit was used (GE Healthcare, Chicago, IL, USA) after staining with monoclonal antibodies by Santa Cruz Biotechnology Inc. (Dallas, TX, USA) or Cell Signaling Technology (Danvers, MA, USA).

### 2.8. Statistical Analysis

All experiments, except for sequencing studies, were performed at least 3 times. To analyze these data, Prism (Version 10.1.0, GraphPad, San Diego, CA, USA) was used applying a one-way ANOVA. Significance was reported when the *p*-value was below 0.05. * <0.05, ** <0.01, *** <0.001, **** <0.0001.

## 3. Results

### 3.1. Midostaurin Inhibits the Generation of Mature DCs

First, we investigated the effect of the three different FLT3i on DC differentiation. After the generation of bmDCs in the presence of different FLT3i or—as a control—DMSO, the proportion of midostaurin-exposed bmDCs, defined as CD11c/CD11b double-positive cells, was significantly reduced in comparison to the control but also to gilteritinib- and quizartinib-exposed bmDCs. Of note, the effect was dose-dependent and most pronounced at a concentration of 100 nM midostaurin (Figure 1a).

In line with these findings, moDCs showed reduced CD1a expression and increased expression for CD14, a classical monocyte marker, at exposure to 100 nM midostaurin (Figure 1b).

### 3.2. FLT3 Inhibitors Do Not Induce Apoptosis in bmDCs

To rule out the toxic effects of FLT3i treatment, the rate of double positive cells for life/dead and Annexin V staining was analyzed. We could show a dose-dependent but not significant effect of the different FLT3 inhibitors on the apoptosis rate of bmDCs, all <20% (Figure 1c).

### 3.3. Midostaurin and Gilteritinib but Not Quizartinib Impair the Costimulatory Function of DCs

The interaction between DCs and T cells is essential for T cell activation. For clonal expansion of naïve T cells, three key signals are essential: antigen presentation by DCs via MHC molecules, upregulation of costimulatory molecules such as CD80 and CD86 on DCs, and the production of cytokines, such as IL-12 [22,23]. For bmDCs, we observed a down-regulation of CD40 and CD86 after midostaurin treatment and, less pronounced but still significant, after gilteritinib treatment, while no changes in the expression levels of costimulatory markers were found after the application of quizartinib (Figure 1d). After DC differentiation in the presence of the different FLT3 inhibitors or DMSO, moDCs were stimulated with LPS to induce maturation. Here, we could detect reduced expression of CD80, CD83 and CD86 after midostaurin treatment, indicating a suppressive effect on moDCs. The effect of gilteritinib was less pronounced. In line with previous effects on bmDC, quizartinib did not heavily impair the expression of costimulatory surface markers (Figure 1e).

### 3.4. Quizartinib, in Contrast to Other FLT3 Inhibitors, Does Not Affect LPS-Induced Cytokine Secretion

In addition to costimulatory molecules, the cytokine environment shapes immune responses [23,32]. Therefore, we analyzed the effect of different FLT3i on the secretion of proinflammatory cytokines after LPS stimulation of murine bmDCs. We detected decreased levels of CCL2 and IL-6 upon treatment with midostaurin and gilteritinib, while only midostaurin treatment reduced the production of IL-12. In contrast, we could not detect any suppressive effects of quizartinib for the tested cytokines (Figure 2a).

In contrast, in human moDCs, all three tested inhibitors dampened IL-6 as well as IL-12 secretion (Figure 2b). Midostaurin and Gilteritinib further reduced TNF-alpha production, which was not altered by quizartinib (Figure 2b).

### 3.5. The 3′-mRNA Analysis Reveals Inhibitory Effects of Midostaurin and Gilteritinib

As previous experiments revealed that (i) midostaurin exerts marked immunosuppressive effects on DCs, (ii) the immunosuppressive potential of gilteritinib is less pronounced and (iii) quizartinib only showed minimal alterations of DC function, we aimed to identify the pathways involved by treatment with FLT3i. Therefore, 3′mRNA was isolated from bmDCs after differentiation in the presence of FLT3 inhibitors (100 nM) and LPS stimulation. By means of hierarchical clustering, we could clearly show the clustering of midostaurin-exposed bmDCs (Figure 3a). Further principal component analysis revealed separated clustering of midostaurin-treated samples, while quizartinib- and gilteritinib-exposed bmDCs showed similar expression profiles as DMSO-treated bmDCs after LPS stimulation. (Figure 3b). Comparing each FLT3 inhibitor treatment with DMSO treatment, midostaurin treatment resulted in a significant differential regulation of more than 800 genes. The following genes were identified as the most upregulated: *Ube2l3*, *Gm2000*, *Ddx6* and *Rnf13*. Gene expression of *Pira1*, *Ifit1bl1*, *Ccl12*, *Cxcl10* and *Ccl12* were amongst the most downregulated (Figure 3c). To investigate the relevance of these gene expression alterations, which are all involved in immune responses, the KEGG pathway analysis was performed and revealed affection of pathways important for proper DC function. The most significantly regulated pathways were the TLR (*p* = 1.87 × 10^−29^; FDR = 9.78 × 10^−28^) and TNF signaling pathways (*p* = 1.90 × 10^−26^; FDR = 5.96 × 10^−25^). Additionally, JAK/STAT, TNF as well as NFκB, MAPK, PI3K, and FoxO signaling were altered (Table 1).

Gene expression alterations after gilteritinib exposure were much more subtle. The highest fold change in gene upregulation was observed for *Socs1*, *Nudt18* and *Pira1*. *Ube2l3*, *Adam9*, *Gm2000* and *Ddx6* showed the highest fold change in the downregulation of genes (Figure 3c). To analyze the biological relevance of these gene dysregulations, the KEGG pathway analysis revealed the most pronounced dysregulation of the JAK-STAT (*p* = 2.91 × 10^−10^; FDR = 1.31 × 10^−8^) and the TNF signaling pathway (*p* = 6.10 × 10^−10^; FDR = 2.39 × 10^−8^) by gilteritinib treatment (Table 2).

Comparing gene expression levels in LPS-maturated and quizartinib-treated bmDCs, we observed the highest fold change in upregulation of the following genes: *Adam9*, *Ube2l3* and *Gm2000* (Figure 3c). No altered pathways were identified by KEGG analysis.

### 3.6. Midostaurin Inhibits Phosphorylation of Stat3, Stat5, Akt and NFκB

To validate our findings obtained from RNA sequencing of murine DCs, we analyzed the expression and phosphorylation of proteins relevant to JAK/STAT, Akt, and the NFκB pathway in human moDCs. Here, we could prove again the suppressive effect of midostaurin and the less pronounced effect of gilteritinib. The expression of phosphorylated Stat3 and Stat5 are clearly impaired by midostaurin (Figure 3d). In line with sequencing data exploiting the PI3K/Akt pathway alteration, phosphorylation of Akt was inhibited by midostaurin and gilteritinib but not quizartinib (Figure 3e). Corroborating our previous results, the inhibition of the NFκB pathway by gilteritinib and midostaurin, and expression of relB and cRel were reduced by these two TKIs (Figure 3f).

## 4. Discussion

Our data clearly identify midostaurin as the FLT3i with the most pronounced suppressive effects on DCs. First, midostaurin treatment reduced the frequency of mature DCs after differentiation from murine bone marrow cells and induced the differentiation of monocytes rather than DCs from human PBMCs characterized by reduced CD1a and increased CD14 expression. We show, by RNA sequencing that TLR and JAK/STAT signaling pathways, TNF signaling, and the NFκB pathway were downregulated by midostaurin exposure, leading to reduced STAT3, STAT5, Akt, cRel and relB phosphorylation. Consequently, midostaurin lowered the expression of B7 molecules and dampened the secretion of pro-inflammatory chemokines and cytokines. These immunosuppressive effects of midostaurin diminish the potential to induce proper T-cell responses via DCs.

Meanwhile, gilteritinib also shows suppressive effects on cytokine release and expression of costimulatory molecules such as CD40, albeit to a lesser extent compared to the effect of midostaurin.

In sharp contrast, we show that quizartinib does not exert any of these strong immunosuppressive effects. Compared to other FLT3i, only slightly reduced levels of IL-6 and IL-12 in human DCs were observed, and RNA sequencing analysis revealed no altered pathways in KEGG analysis.

The discovery of FLT3 as a therapeutic target in AML has improved patient prognosis [3], and different TKIs have shown activity in FTL3mut AML. As midostaurin is a multi-targeted TKI with various other inhibitory effects, gilteritinib and quizartinib have greater specificity for FLT3. Of note, quizartinib is only effective in ITD mutations as it binds only the inactive form of the FLT3 receptor, while both other TKIs bind the FLT3 receptor in the active confirmation [4]. Both the binding of only the inactive form and the greater specificity for FLT3 might explain the lack of immunosuppressive activity of quizartinib. 

However, as responses may be of limited duration, consolidation therapy is mandatory, and alloHSCT is the post-induction therapy of choice for suitable patients with FLT3_pos_ AML.

Post-alloHSCT FLT3-directed therapy with sorafenib improves patient outcomes [9,33]. Data regarding the novel FLT3i midostaurin, gilteritinib and quizartinib are evolving, but the best choice for maintenance therapy after alloHSCT is yet to be determined.

Midostaurin used as maintenance therapy after alloHSCT shows prolonged overall survival [34]. Rates of acute GvHD grade 2 to 3 were slightly decreased (standard of care vs. midostaurin 37% vs. 30%) as well as the incidence of chronic GvHD (47% for standard of care and 37% for midostaurin) [11]. Of interest, midostaurin treatment was associated with a high incidence of invasive fungal disease [35]. Our data support this observation, as we see a profound inhibition of DC function by the effects of midostaurin. The investigated surface markers [22,36,37] and TNFα [32], IL-6 [38], IL-12 [23], and CCL2 [39] are essential to induce proper T cell-mediated immune responses and foster Th1/Th2 differentiation. Mechanistically, we could show that midostaurin impaired several relevant signaling pathways, amongst them, TLR, TNF, and NFκB signaling and subsequent cytokine secretion of IL-6 and IL-12 are essential for fungal immunity [40].

Gilteritinib maintenance therapy, tested in the relapsed or refractory setting, resulted in similar GvHD rates after alloHSCT compared to midostaurin (33%) [41]. Pyrexia (43%) and pneumonia (25%) were common side effects of the therapy. The MORPHO trial, the only phase III trial comparing gilteritinib vs. placebo as maintenance after alloHSCT, only showed survival benefits for FLT3_pos_ patients with detectable minimal residual disease after alloHSCT [42]. GvHD rates were not statistically affected by gilteritinib treatment, but infectious complications were increased. Our data might partly explain these observations as pro-inflammatory pathways, and cytokine secretion in DCs were inhibited by gilteritinib. Of note, not only effects on DCs may contribute to this observation but also myelosuppression [42].

In line with other approaches for post-transplant FTL3 targeted therapy, a study investigating quizartinib maintenance showed prolonged survival [43]. Sadly, GvHD rates were not reported due to a lack of post-transplant data. However, another phase 1 study with 13 patients reported similar GvHD rates as for the other FLT3i mentioned above (23% grade 2 and 8% grade 3). However, grade 3 pneumonia was only reported in 8% of patients, corroborating the results of other studies showing low rates of febrile neutropenia under quizartinib exposure [44]. These clinical observations align with our results, as we see only minor immunosuppression by quizartinib.

A trial with head-to-head comparison between these three FLT3i is lacking. Regarding side effects, our data corroborate clinical observations, as midostaurin has the most pronounced immunosuppressive effect resulting in slightly decreased GvHD rates while favoring fungal infections. Less pronounced, gilteritinib showed inhibitory effects on DC function with increased infection rates in the MORPHO trial. In contrast, infection rates under quizartinib treatment appear to be low, which might be partly explained by the missing inhibition of pro-inflammatory pathways in DCs.

However, there are limitations to our study, as we did not investigate the effect of FLT3i in patient samples. Moreover, treatment with FLTi induces myelosuppression and decreased numbers of neutrophil granulocytes and not only DCs but also T and B cell functions have to be taken into account when investigating immune responses. Additionally, when choosing the most suitable compound for maintenance therapy, it has to be taken into account that quizartinib only inhibits FLT3 ITD AML, while midostaurin and gilteritinib also target FLT3 TKD.

## 5. Conclusions

We show in different approaches the pleiotropic suppressive effects on DCs of the FLT3i midostaurin and, less pronounced, gilteritinib in comparison to quizartinib. The results of our study may explain the increased rate of fungal infections in midostaurin-treated patients. Therefore, our study may provide an additional rationale to optimize the selection of FLT3i for post-alloHSCT maintenance therapy.

## Figures and Tables

**Figure 1 cancers-16-03719-f001:**
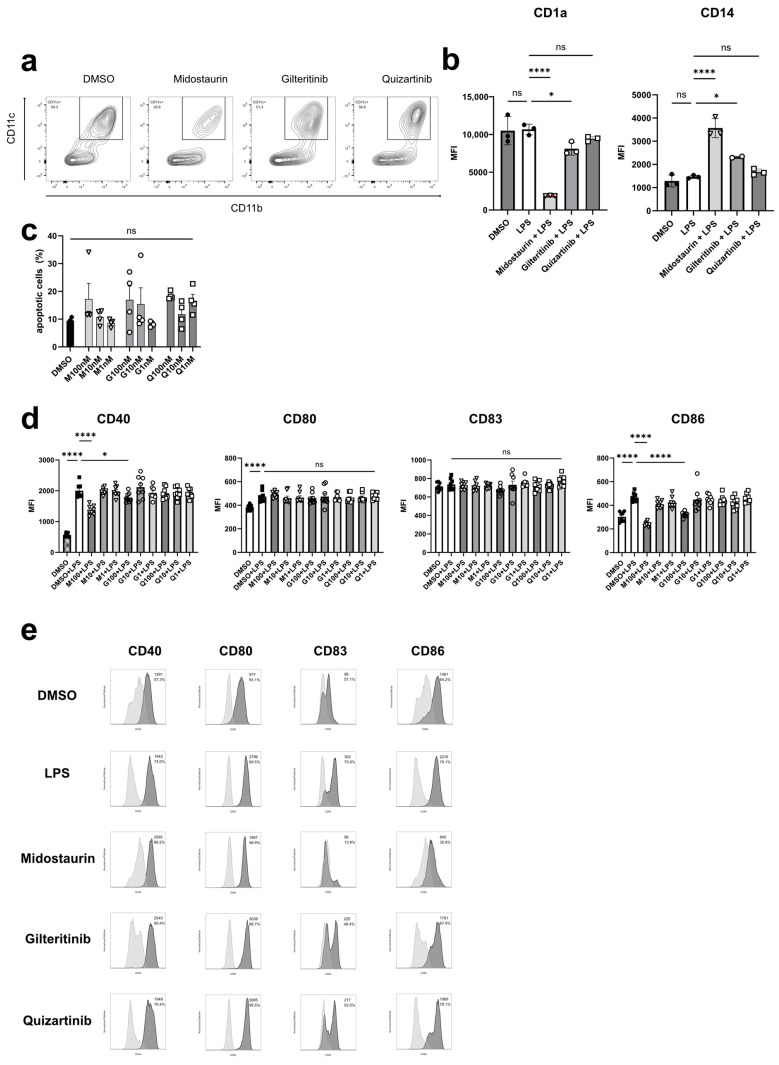
Effect of FLT3 inhibition on DC differentiation and surface expression of costimulatory molecules. DCs generated from murine bone marrow cells were differentiated in the presence of 100 nM gilteritinib, midostaurin or quizartinib and were gated using forward and sideward scatter. Dead cells were excluded by life/dead staining. Proportion of double-positive CD11c+/CD11b+ cells is markedly reduced after differentiation in the presence of midostaurin (**a**). Human PBMCs were cultured to obtain DCs as described. Midostaurin-induced monocyte differentiation indicated by reduced CD1a and increased CD14 expression in the presence of midostaurin (*p* < 0.0001) and gilteritinib (*p* < 0.05) (**b**). Differentiation of murine bone marrow cells in the presence of midostaurin (M1 = 1 nM; M10 = 10 nM; M100 = 100 nM), gilteritinib (G1 = 1 nM; G10 = 10 nM; G100 = 100 nM) or quizartinib (Q1 = 1 nM; Q10 = 10 nM; Q100 = 100 nM) in different concentrations did not induce apoptosis (**c**). The mean fluorescence intensity of CD40 and CD86 on bmDCs after LPS stimulation was reduced in a dose-dependent manner by midostaurin (*p* < 0.0001) and gilteritinib (*p* < 0.05) exposure (**d**). MoDCs showed reduced CD80, CD83 and CD86 expression after midostaurin treatment with 100 nM. The effect was less pronounced after gilteritinib (100 nM) and only minimal after quizartinib (100 nM) treatment. MFI and rates of positive cells are depicted in the right upper corner (**e**). ns not significant, * *p* < 0.05, **** = *p* < 0.0001.

**Figure 2 cancers-16-03719-f002:**
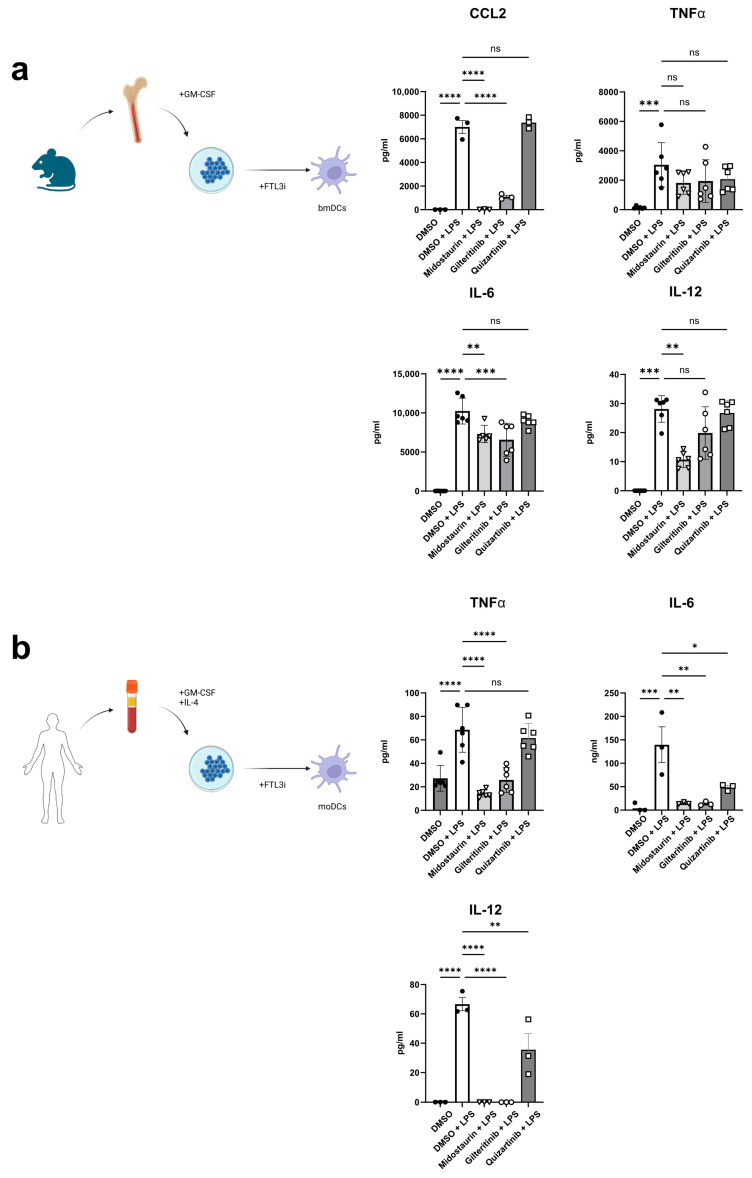
FTL3 inhibition suppresses cytokine secretion by dendritic cells. After 6 days of differentiation in the presence of FLT3i at 100 nM concentration, cytokine release of bmDCs was measured 18 h after LPS stimulation. IL-12 levels were only reduced after midostaurin treatment (*p* < 0.01), while IL-6 (*p* < 0.01 for midostaurin and *p* < 0.001 for gilteritinib) and CCL-2 levels (*p* < 0.0001 for both FLT3i) were also negatively affected by gilteritinib treatment. TNFα secretion was not affected. The generation of bmDCs is illustrated. Created with BioRender.com/j07f023 (**a**). For moDCs, midostaurin (100 nM) and gilteritinib (100 nM) treatment reduces TNFα (*p* < 0.0001), IL-6 (*p* < 0.01) and IL-12 levels (*p* < 0.0001), quizartinib (100 nM) treatment did not affect TNFα levels but lowered IL-12 (*p* < 0.01) and IL-6 (*p* < 0.05) levels. Schematic illustration of moDC generation created with BioRender.com/u60g276 (**b**). ns not significant, * = *p* < 0.05, ** = *p* < 0.01, *** = *p* < 0.001, **** = *p* < 0.0001.

**Figure 3 cancers-16-03719-f003:**
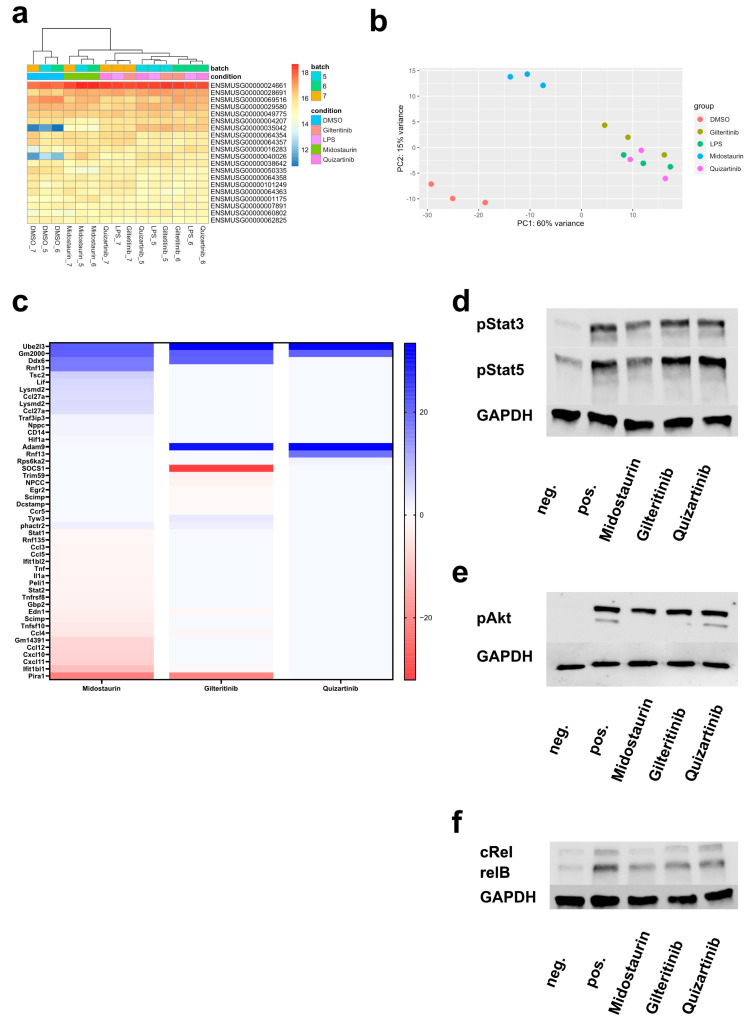
RNA sequencing reveals inhibition of important immune response pathways by FLT3i, which is supported by protein expression analysis. BmDCs were cultured in the presence of DMSO or FLT3i. After 6 days, cells were stimulated with LPS, when indicated, and RNA was harvested 18 h later. Using hierarchical clustering, the top 20 differentially expressed genes found in RNA sequencing analysis reveal specific clustering of DMSO control samples and after exposure to midostaurin but not for other treatment groups. Expression values are depicted from low (blue) to high (red) (**a**). Concomitantly, principal component analysis (PCA) based on all analyzed gene sets also showed high similarity of midaustaurin-treated samples (**b**). Log-fold change in gene expression of midostaurin-, gilteritinib- and quizartinib-treated moDCs after LPS maturation is depicted in a heatmap. Up- (blue) and downregulation (red) are color-coded (**c**). To prove the inhibition of relevant pathways, protein was extracted from human moDCs after FLT3i treatment and LPS stimulation. Stat3 and Stat5 phosphorylation was impaired by midostaurin treatment (**d**). Inhibition of Akt phosphorylation was observed after midostaurin and quizartinib treatment (**e**). The NFκB pathway was inhibited by midostaurin and gilteritinib treatment as cRel and relB expression were reduced. Quizartinib did not affect relB or cRel expression (**f**). The uncropped bolts are shown in Appendix A.

**Table 1 cancers-16-03719-t001:** KEGG analysis of midostaurin-treated bmDCs (selected pathways).

Pathway	Total	Expected	Hits	*p* Value	FDR
Toll-like receptor signaling pathway	99	9.6	54	1.87 × 10^−29^	9.78 × 10^−28^
TNF signaling pathway	110	10.7	54	1.90 × 10^−26^	5.96 × 10^−25^
Th17 cell differentiation	102	9.89	49	1.47 × 10^−23^	3.30 × 10^−22^
IL-17 signaling pathway	91	8.82	46	2.22 × 10^−23^	4.64 × 10^−22^
MAPK signaling pathway	294	28.5	82	1.17 × 10^−19^	2.16 × 10^−18^
PI3K-Akt signaling pathway	358	34.7	92	2.61 × 10^−19^	4.32 × 10^−18^
FoxO signaling pathway	132	12.8	51	2.97 × 10^−19^	4.66 × 10^−18^
NOD-like receptor signaling pathway	206	20	65	8.75 × 10^−19^	1.19 × 10^−17^
NF-kappa B signaling pathway	109	10.6	44	1.25 × 10^−17^	1.63 × 10^−16^
Neurotrophin signaling pathway	121	11.7	44	1.27 × 10^−15^	1.21 × 10^−14^
ErbB signaling pathway	84	8.14	35	8.76 × 10^−15^	6.97 × 10^−14^
Jak-STAT signaling pathway	165	16	51	1.44 × 10^−14^	1.08 × 10^−13^
RIG-I-like receptor signaling pathway	68	6.59	31	1.49 × 10^−14^	1.09 × 10^−13^
HIF-1 signaling pathway	105	10.2	39	2.43 × 10^−14^	1.73 × 10^−13^
B cell receptor signaling pathway	72	6.98	31	1.03 × 10^−13^	6.90 × 10^−13^
T-cell receptor signaling pathway	101	9.79	36	1.04 × 10^−12^	6.53 × 10^−12^
Chemokine signaling pathway	200	19.4	53	4.16 × 10^−12^	2.56 × 10^−11^
Ras signaling pathway	233	22.6	57	2.17 × 10^−11^	1.24 × 10^−10^
Cytokine-cytokine receptor interaction	296	28.7	63	9.85 × 10^−10^	4.91 × 10^−9^
AMPK signaling pathway	126	12.2	33	6.83 × 10^−8^	2.65 × 10^−7^
Th1 and Th2 cell differentiation	87	8.44	26	9.83 × 10^−8^	3.72 × 10^−7^
Natural killer cell-mediated cytotoxicity	118	11.4	31	1.58 × 10^−7^	5.84 × 10^−7^
VEGF signaling pathway	58	5.62	20	2.24 × 10^−7^	7.98 × 10^−7^
TGF-beta signaling pathway	93	9.02	23	1.85 × 10^−5^	5.62 × 10^−5^

**Table 2 cancers-16-03719-t002:** KEGG analysis of gilteritinib-treated bmDCs (selected pathways).

Pathway	Total	Expected	Hits	*p* Value	FDR
Jak-STAT signaling pathway	165	1.28	13	2.91 × 10^−10^	1.31 × 10^−8^
TNF signaling pathway	110	0.854	11	6.10 × 10^−10^	2.39 × 10^−8^
IL-17 signaling pathway	91	0.707	10	1.55 × 10^−9^	5.39 × 10^−8^
MAPK signaling pathway	294	2.28	14	3.96 × 10^−8^	1.04 × 10^−6^
Toll-like receptor signaling pathway	99	0.769	8	8.40 × 10^−7^	1.46 × 10^−5^
Th17 cell differentiation	102	0.792	8	1.06 × 10^−6^	1.66 × 10^−5^
HIF-1 signaling pathway	105	0.815	8	1.32 × 10^−6^	1.88 × 10^−5^
Th1 and Th2 cell differentiation	87	0.676	7	4.41 × 10^−6^	5.54 × 10^−5^
T-cell receptor signaling pathway	101	0.784	7	1.19 × 10^−5^	0.000121
B cell receptor signaling pathway	72	0.559	6	1.82 × 10^−5^	0.000171
Neurotrophin signaling pathway	121	0.94	7	3.87 × 10^−5^	0.00032
Cytokine-cytokine receptor interaction	296	2.3	10	8.45 × 10^−5^	0.000603
PI3K-Akt signaling pathway	358	2.78	10	0.000398	0.0024
ErbB signaling pathway	84	0.652	5	0.000466	0.00266
FoxO signaling pathway	132	1.03	6	0.000531	0.00287
NF-kappa B signaling pathway	109	0.847	5	0.00152	0.00723
RIG-I-like receptor signaling pathway	68	0.528	4	0.00186	0.00823
Ras signaling pathway	233	1.81	7	0.00209	0.00901
Chemokine signaling pathway	200	1.55	6	0.00443	0.0186
TGF-beta signaling pathway	93	0.722	4	0.00578	0.0239

## Data Availability

The raw data supporting the conclusions of this article will be made available by the authors upon request.

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
