# Peer review of "The Immunomodulatory Effect of Different FLT3 Inhibitors on Dendritic Cells"

_cancers, 2024, doi:10.3390/cancers16213719_

Round 1

Reviewer 1 Report

Comments and Suggestions for Authors

This is a very interesting paper on a topic with biological and clinical relevance in FLT3-mutated AML. The main message (mido is immunosuppressive, gilt less immunosuppressive and quiz has almost no immunosuppressive effect) is well supported by data and in part intuitituve given the large mido spectrum of activity and the more targetd activity of the other two drugs. However, is counter intuitive why quiz has almost no effect even in the presence of clinically documented anti-FLT3 activity. To help readers I suggest to dedicate a section of the discussion chapter on a brief description of the spectrum of TKI activities of the 3 molecules and of the putative reason of the lack of immunosuppressive activity of quiz. 

Author Response

[Comments 1] To help readers I suggest to dedicate a section of the discussion chapter on a brief description of the spectrum of TKI activities of the 3 molecules and of the putative reason of the lack of immunosuppressive activity of quiz. 

[Response 1] Thank you for the valuable comment. We agree that a short passage might improve the manuscript and added the following lines: “and different TKIs have shown activity in FTL3mut AML. As midostraurin is a multi-targeted TKI with various other inhibitory effects, gilteritinib and quizartinib have greater specificity for FLT3. Of note, quizartinib is only effective in ITD mutations as it binds only the inactive form of the FLT3 receptor, while both other TKIs bind the FLT3 receptor in the active confirmation[4]. Both, binding of only the inactive form and the greater specificity for FLT3, might explain the lack of immunosuppressive activity of quizartinib.”

Reviewer 2 Report

Comments and Suggestions for Authors

This is an interesting study, which aims to determine 3 different tyrosine kinase inhibitors’ (TKIs) inhibitory effect on normal murine or human DC cells. Based on their experimental results, they hypothesize that increased infection rates can be caused by TKI’s inhibitory role in DC cells in AML patients.  

I have following comments: 

1.     Please add “normal” before dendritic cells in vitro in Simple Summary section. 

2.     For 2.1 Methods: Please describe the exact date of adding TKI, for how many days of TKI treatment before the LPS activation.  Also, please explain the reason of the current treatment order, why not add LPS first to activate DC, then treat them with TKI?

3.     Because authors used normal DC, the RNA-seq results and limited panel of cytokine secretion are not informative. Can authors examine the effect of different TKI on DCs from AML patient samples? 

Author Response

[Comments 1]Please add “normal” before dendritic cells in vitro in Simple Summary section. 

[Response 1] Thank you for this comment. To increase the comprehensiveness of the simple summary we included “normal” as suggested.

[Comments 2] For 2.1 Methods: Please describe the exact date of adding TKI, for how many days of TKI treatment before the LPS activation.  Also, please explain the reason of the current treatment order, why not add LPS first to activate DC, then treat them with TKI?

[Response 2] We agree with this comment. To be more precise, we added the underlined text to the method section: midostaurin, gilteritinib and quizartinib (all purchased from Selleckchem) were added at specific concentrations ranging from 1 to 100 nM every other day for the whole differentiation period of 6 days to mimic clinical routine, where continuous TKI treatment affects DC differentiation and development as well. We chose the described treatment order as it reflects the in vivo condition in a better way, as patients take these drugs continuously and therefore a continuous drug exposure in vitro reflects the clinical routine more precisely.

[Comments 3]Because authors used normal DC, the RNA-seq results and limited panel of cytokine secretion are not informative. Can authors examine the effect of different TKI on DCs from AML patient samples? 

[Response 3] Thank you for this valuable comment. As stated in our discussion, our study is limited as we did not have the opportunity to investigate patient samples. Of note, the analysis of patient samples would be very interesting but has also some restrictions as the number of circulating DCs is quite low and differentiation of TKI-treated PBMCs into moDCs would be artificial as well.  Of course, DCs from AML patients may also harbor FLT3 mutations and therefore the investigation of patient samples may be of interest, however we have to consider that in clinical routine the median time to response is quite short, for example with 2.3 months for gilteritinib monotherapy (Perl et al., NEJM, 2019) and even shorter in combination therapy with anthracyclines and cytarbine. After that, the malignant FLT3mut hematopoiesis is no longer present and therefore we believe it is of greater relevance to investigate the effect of FLT3 inhibitors on “normal” DCs. Even for patients after alloHSCT, who receive FLT3-directed therapies as maintenance therapy, there is no FLT3 mutant hematopoiesis and the effect of FTL3 inhibition is restricted to “normal” DCs. Taken all these points together, we are convinced that the analysis of “normal” DCs is providing valuable information.